# Production of Feline Universal Erythrocytes with Methoxy Polyethylene Glycol

**DOI:** 10.3390/jfb14090476

**Published:** 2023-09-18

**Authors:** Hyung Kyu Kim, Dan Bi Ahn, Han Byeol Jang, Jing Ma, Juping Xing, Joo Won Yoon, Kyung Hee Lee, Dong Min Lee, Chang Hyun Kim, Hee Young Kim

**Affiliations:** 1Department of Physiology, Yonsei University College of Medicine, Seoul 03722, Republic of Korea; badawanabi@knu.ac.kr (H.K.K.); anda1434@yuhs.ac (D.B.A.); star961018@naver.com (H.B.J.); majing981018@naver.com (J.M.); xingjuping@yonsei.ac.kr (J.X.); joowonyoon77@gmail.com (J.W.Y.); 2Department of Oral Physiology, School of Dentistry, Kyungpook National University, Daegu 41940, Republic of Korea; 3KABB, Daegu 42212, Republic of Korea; lkh277@hanmail.net (K.H.L.); mutalisk2@naver.com (D.M.L.); agoni@naver.com (C.H.K.)

**Keywords:** mPEG, universal red blood cells, RBC, feline

## Abstract

Blood group mismatch in veterinary medicine is a significant problem in blood transfusion, sometimes leading to severe transfusion reactions and even patient death. Blood groups vary from species to species and there are three known blood groups in cats: A, B and AB. While A-type cats are most common, there is a shortage of feline B-type blood groups in cats. By using methoxy polyethylene glycol (mPEG) to protect antigenic epitopes on red blood cells (RBCs), we aimed to find the optimal conditions for the production of feline universal RBCs. The surfaces of feline A-type RBCs were treated with mPEG at various molecular weights and concentrations. Agglutination tests showed that the coating of feline A-type RBCs with mPEG of 20 kDa and 2 mM blocked hemagglutination to feline anti-A alloantibodies over 8 h. While no differences in RBC size and shape between intact and mPEG-treated RBCs were seen, coating RBCs with mPEG inhibited the binding of feline anti-A alloantibodies. Furthermore, the mPEG-treated RBCs did not cause spontaneous hemolysis or osmotic fragility, compared to control RBCs. According to a monocyte monolayer assay, mPEG treatment significantly reduced feline anti-A antibody-mediated phagocystosis of RBCs. These results confirm the potential of using activated mPEG on feline A-type RBC to create universal erythrocytes for transfusion to B-type cats.

## 1. Introduction

Blood transfusion is a life-saving treatment for severe anemia and coagulopathies that is continually increasing. In veterinary medicine, blood transfusions are infrequently performed in primary care veterinary practice and have become an important component of intensive medical care [1]. While potentially a lifesaving procedure, the blood transfusion can be detrimental to the recipient if performed without precautions (i.e., mismatched blood transfusions).

Transfusion of incompatible bloods is a significant problem in veterinary medicine, sometimes leading to severe transfusion reactions and even patient death [2]. Blood groups vary from species to species. While blood groups in dogs and cats have been studied for more than half a century, their biochemical and molecular genetic characteristics have not been fully elucidated [3,4]. Additional blood groups in dogs and cats have been newly discovered, including canine Dal type, feline Mik type, and other unclassified blood types [5,6,7,8]. There are three known blood groups in cats: A, B and AB. In particular, B-type cats have naturally occurring strong antibodies to A antigens, causing fatal hemolytic transfusion reactions against transfused A-type bloods. While A-type cats are the most common, B-type cats are quite rare. For example, 1% of the feline population in Japan, 0.3~1.7% in the USA, 2.9% in Scotland, and 0.4% in Switzerland are B-type cats. This may cause a shortage of B-type bloods for emergency blood transfusion in anemic B-type cats [9]. Currently, no satisfactory solutions exist to replace feline B-type bloods.

Many attempts have been made to produce red blood cells (RBCs) that can be transfused regardless of human ABO group of the donor and recipient. The approach involves masking the A and B antigens and all other blood group antigens with polyethylene glycol (PEG), yielding so-called stealth RBCs or universal RBCs; the hope is that such RBCs will not react with any blood group antibodies and may not be recognized as foreign, thus not initiating an immune response [10]. Dr. Scott et al. reported that masking human RBCs with methoxy PEG (mPEG) did not affect their structure and ability to carry oxygen, and that mPEG-treated human RBCs showed diminished immune responses in mice [11].

To create feline universal RBCs, we explored the optimal conditions of mPEG for feline A-type bloods, the morphology and stability of mPEG-treated feline RBCs, and in vitro phagocytosis with feline universal RBCs.

## 2. Materials and Methods

### 2.1. Chemicals and Reagents

Methoxy PEG-succinimidyl valerate (mPEG-SVA; molecular weight: 2 kDa, 5 kDa, 10 kDa, 20 kDa, 30 kDa; Laysan bio, Inc., Arab, AL, USA), mPEG-SVA 40 kDa (Advanced BioChemicals, Lawrenceville, GA, USA), feline anti-A alloantibodies (with a titer 1:8; KABB, Suseong-gu, Daegu, Republic of Korea), and goat anti-cat IgG-FITC (Alexa Fluor 488, Abcam, Trumpington, Cambridge, UK) were used.

### 2.2. Feline Blood Samples

All procedures were approved by the Institutional Animal Care and Use Committee (IACUC) of Yonsei University and Daegu Haany University and conducted according to the National Institutes of Health Guide for the Care and Use of Laboratory Animals. Feline A-type whole bloods in citrate-phosphate-dextrose adenine (CPDA-1 containing dextrose, sodium citrate, citric acid, monobasic sodium phosphate and adenine) anticoagulant were donated from the Korea Animal Blood Bank (KABB, Suseong-gu, Daegu, Republic of Korea) and stored at 1~6 °C until used. To prevent any storage artifacts, fresh whole blood less than 7 days from collection was used. The feline anti-A alloantibodies were heated for 30 min at 56 °C to inactivate the complement, aliquoted, and frozen at −80 °C until use.

### 2.3. PEGylation of Cell Surface

Treatment of mPEG on feline RBCs was carried out as described previously [11]. The molecular weights of mPEG were chosen based on previous human RBC studies [12]. In brief, whole blood was first centrifuged to discard plasma and then washed 3 times with saline at 3000 rpm for 5 min. Prior to use, mPEGs of various molecular weights (2, 5, 10, 20, 30 or 40 kDa) were dissolved at concentrations of 0.1, 0.2, 0.5, 1, 2, 5 or 10 mM in isotonic alkaline phosphate-buffered saline (PBS; 105 mM NaCl, 50 mM K_2_HPO_4_, pH 8.0) and added to RBCs with a hematocrit of about 10%. The RBCs were incubated for 30 min at 25 °C, washed 3 times with saline and suspended to a hematocrit of 50% in saline.

### 2.4. Agglutination Assay

A direct agglutination test was performed as previously described with slight modifications [13]. In brief, 50 μL of the control and mPEG-treated RBCs (50% in saline) were placed in 24-well plates. Then, 170 μL of feline anti-A alloantibodies (1:8 tires) was added to each well, and the cells were incubated for 15, 60, 120, 180, 240, 300 and 480 min at room temperature. Incubated cells were stirred at each time point, observed, and imaged using a digital camera. In qualitative assessments, agglutination intensity was graded on a scale from 0 ~ to 3+ as described previously [13] as follows: 0, negative; 1, small clumps with ~25% agglutination; 2, ~50% agglutination; and 3, more than 75% agglutination or complete agglutination.

The microcolumn gel card method (DG Gel, Grifols, Italy) was also carried out, as described previously [14,15]. In brief, 30 μL of the blood samples (10% in saline) was mixed with 70 μL of the feline anti-A alloantibodies (1:8 tires), followed by 15 min incubation at room temperature (RT). Gel columns were centrifuged for 10 min at 80× *g* in the manufacturer’s centrifuge. The RBC retention in the gel was graded as follows: 0, all RBCs at the bottom of gel; 1+, few RBC agglutinates in the lower half of gel but most RBCs at the bottom of gel; 2+, RBC agglutinates throughout gel and RBCs on upper surface; and 3+, all RBCs on top of gel. An RBC retention of ≥2+ was considered positive.

### 2.5. Morphology of mPEG-Treated RBCs and Immunofluorescence of mPEG-Treated RBCs Incubated with Feline Anti-A Alloantibodies

To examine the morphological characteristics of RBCs, control (intact) or mPEG-treated RBCs were diluted in PBS (1:1000). Blood samples (10 μL per slide) were placed on adhesive microscope slides and covered with coverslips. The slides were then imaged under a bright field microscope (Olympus BX41, Shinjuku, Tokyo, Japan) at 1000× magnification.

For the immunofluorescence study, the control (intact) or mPEG-treated RBCs were incubated with feline anti-A alloantibodies (1:8 titre) at room temperature for 1 h. After washing, the cells were incubated with goat anti-cat IgG-FITC for 1 h at room temperature. The cells were then washed three times with saline and diluted in PBS (1:1000). The diluted cells were placed on adhesive microscope slides and imaged under a fluorescence microscope (Olympus BX41, Japan).

### 2.6. Measurement of Hemolysis

The spontaneous hemolysis of PBS-treated (*n* = 3) or mPEG-treated (*n* = 3) RBCs was quantified using Drabkin’s method, as previously described (Moore, Ledford, and Merydith 1981). Briefly, the cells were incubated for 0, 24, or 48 h at 4 °C in Eppendorf tubes. After the incubation, the samples were centrifuged at 3000× *g* for 5 min, and the supernatants were transferred and then incubated with Drabkin’s reagent for 15 min at room temperature. Concentrations of free hemoglobin were measured at 540 nm using a microplate reader (Multiskan SKY, Thermo Fisher Scientific, Waltham, MA, USA).

### 2.7. Osmotic Fragility Test

Osmotic fragility was determined as previously described [16]. In brief, 10 μL of PBS-treated (*n* = 9) or mPEG (2 mM, 20 kDa)-treated RBCs (hematocrit = 50%) were suspended in 1 mL NaCl solution with increasing concentrations (0.000, 0.150, 0.250, 0.275, 0.300, 0.350, 0.375, 0.400, 0.425, 0.475, 0.500, 0.550, 0.650, and 0.900%) and incubated for 60 min at 37 °C. After centrifuging the samples at 3000× *g* for 5 min at room temperature, the hemoglobin concentration in the supernatant was measured using Drabkin’s method (Moore, Ledford, and Merydith 1981).

### 2.8. Monocyte Monolayer Assay (MMA)

The monocyte monolayer assay was conducted as described with slight modifications (Li et al., 2015). Peripheral blood mononuclear cells (PBMCs) were isolated from feline whole blood via density gradient centrifugation. The PBMCs were then diluted in RPMI 1640 complete medium and seeded on 8-chamber slides (Thermo Scientific, Waltham, MA, USA). Cells were cultured overnight. RBCs (either PBS- or mPEG-treated, ~75 µL of a 3% hematocrit) were mixed with anti-A plasma and incubated at 37 °C for 1 h. The medium was discarded from the incubated slides to remove unattached cells. The opsonized RBCs were washed, diluted in RPMI complete medium, and seeded onto monocyte monolayers. The cells were then incubated at 37 °C for 2 h. After incubation, the slides were washed, fixed in 100% methanol, and stained with Wright–Giemsa. Phagocytosis was evaluated under light microscopy. The monocyte index was calculated as the percentage of macrophages with adherent or phagocytized erythrocytes in total macrophages.

### 2.9. Statistical Analysis

Data were presented as mean ± standard error of the mean (SEM) and two-way measurement analysis of variance (ANOVA), followed by post hoc testing using Tukey’s method, or paired t-test, where appropriate. Statistical significance was considered at *p* < 0.05.

## 3. Results

### 3.1. Effect of mPEG on Antigenic Recognition

The antigenicity of mPEG-treated feline A-type RBCs was assessed by a direct agglutination assay following an incubation with feline anti-A alloantibodies. Control and mPEG-treated RBCs were incubated for 15, 60, 120, 180, 240, 300 and 480 min after the addition of feline anti-A alloantibodies (Figure 1A). Agglutination intensity was graded on a scale from 0 ~ to 3+ on the time points (Figure 1B). The RBCs were conjugated with mPEG of various molecular weights (2, 5, 10, 20, 30 and 40 kDa) at concentrations of 0.1, 0.2, 0.5, 1, 2, 5 or 10 mM. With 15 min incubation, treatment with mPEG of 10, 20, 30 and 40 kDa completely prevented agglutination at concentrations of 2, 5 and 10 mM, while mPEG of 2 kDa and 5 kDa failed to prevent the agglutination (two-way ANOVA; 15 min; group F_(6,168)_ = 120.698, *p* < 0.001; concentration F_(7,168)_ = 273.221, *p* < 0.001; interaction F_(42,168)_ = 16.291, *p* < 0.001; Figure 1C). When the mPEG-treated RBCs were examined after incubation for 60, 120 and 180 min, complete prevention of agglutination was observed at concentrations of 5 and 10 mM in the mPEG groups of 10, 20, 30 and 40 kDa, but neither 2 nor 5 kDa (two-way ANOVA; 60 min; group factor F_(6,168)_ = 140.8, *p* < 0.001; concentration factor F_(7,168)_ = 286.397, *p* < 0.001; interaction F_(42,168)_ = 19.595, *p* < 0.001; 120 min; group factor F_(6,168)_ = 175.932, *p* < 0.001; concentration factor F_(7,168)_ = 359.825, *p* < 0.001; interaction F_(42,168)_ = 26.652, *p* < 0.001; 180 min; group factor F_(6,168)_ = 192.186, *p* < 0.001; concentration factor F_(7,168)_ = 368.415, *p* < 0.001; interaction F_(42,168)_= 28.771, *p* < 0.001; Figure 1D–F). On the time points of 240, 300 and 480 min incubation, the mPEG groups of 20, 30 and 40 kDa blocked agglutination at the concentration of 2, 5 and 10 mM. It was noted that the mPEG treatment of 30 and 40 kDa at high concentrations of 5 and 10 mM revealed slight hemolysis in the supernatants (240 min; group factor F_(6,168)_ = 200.25, *p* < 0.001; concentration factor F_(7,168 )_ = 386.875, *p* < 0.001; interaction F_(42,168)_ = 29.208, *p* < 0.001; 300 min; group factor F_(6,168)_ = 210.276, *p* < 0.001; concentration factor F_(7,168)_ = 417.333, *p* < 0.001; interaction F_(42,168)_ = 31.632, *p* < 0.001; 480 min; group factor F_(6,168)_ = 213.061, *p* < 0.001; concentration factor F_(7,168)_= 441.952, *p* < 0.001; interaction F_(42,168)_ = 31.776, *p* < 0.001; Figure 1G–I). Based on these findings, mPEG of 20 kDa at 2 mM was used for subsequent experiments.

To further confirm whether mPEG treatment ablates the antigenicity of feline A-type RBCs, the experiment was repeated using the microcolumn gel card method. The mPEG-treated feline A-type RBCs were mixed with feline anti-A alloantibodies (Anti-A). Gel columns were centrifuged for 10 min at 80× *g* in the manufacturer’s centrifuge. After incubation and centrifugation, the RBC retention in the gel was graded as 0 (negative) ~3+ (severe agglutination) (Figure 2A). Treatment with mPEG of 2 and 5 kDa molecules at 0.1~10 mM showed all theRBCs on top of the gel (3+ agglutination), and mPEG of 10 kDa molecules also revealed moderate agglutination (1+~2+) of RBC throughout the gel and RBCs on the upper surface, indicating agglutination of the engineered RBCs. On the other hand, mPEG treatment of 20 kDa at concentrations of 2, 5 and 10 mM showed almost negative agglutination (wo-way ANOVA; group F_(5,144)_ = 104.285, *p* < 0.001; concentration F_(7,144)_ = 50.221, *p* < 0.001; interaction F_(35,144)_ = 9.856, *p* < 0.001; Figure 2B,C). Although 30 and 40 kDa also blocked the RBC agglutination up to 5 mM, hemolysis was found in high doses of 10 mM. In Figure 2B, one of four samples at 10 mM displayed 2+ agglutination, and this caused greater agglutination scores than those at 5 mM. In summary, while the 2 and 5 kDa polymer sizes failed to reduce agglutination, the 20, 30 and 40 kDa polymers effectively prevented alloantibody-mediated agglutination at dosages over 0.5 mM.

### 3.2. Morphology of mPEG-Treated RBCs and Inhibition of Antibody Binding by mPEG-Coated RBCs

To see the morphological changes in mPEG-treated RBCs, feline A-type RBCs were treated with mPEG of 20 kDa at 2 mM and compared with control (intact) RBCs under a bright field microscope. The mPEG-treated RBCs showed no morphological alterations in their biconcave shape and color compared to intact RBCs (Figure 3A,B). There was no significant difference in RBC size between the control and mPEG-treated RBCs (*p* = 0.435; Figure 3C).

To identify whether mPEG treatment of RBCs inhibits antibody binding to feline A-type RBCs, feline A-type RBCs coated with mPEG of 20 kDa at 2 mM or control feline A-type RBCs were incubated with the primary feline anti-A alloantibodies and secondary green fluorescent goat anti-cat IgG-FITC. While control RBCs showed strong green fluorescence, no apparent green fluorescence was observed in the mPEG-coated RBCs (* *p* < 0.001; Figure 3D–F). This indicates that the mPEG of 20 kDa at 2 mM effectively blocked the antigenicity of A-type RBCs.

### 3.3. Physiochemical Properties of mPEG-Treated RBCs

To see whether mPEG treatment of RBCs alters physiochemical properties, spontaneous hemolysis and osmotic fragility were compared between control (intact) and mPEG-treated RBCs. For spontaneous hemolysis, feline A-type RBCs coated with mPEG of 20 kDa at 2 mM or control feline A-type RBCs were incubated for 0, 24 or 48 h at 4 °C. While hemolysis was determined immediately, 24, and 48 h after the PEGylation of RBCs, there were no significant differences in the extent of hemolysis between the control and mPEG-treated RBCs (two-way RM ANOVA; group F_(2,12)_ = 104.285, *p* = 1.766; concentration F_(1,12)_ = 50.221, *p* = 0.0741; interaction F_(2,12)_ = 9.856, *p* = 0.694; Figure 4A).

For osmotic fragility, erythrocyte resistance to hemolysis was measured while being exposed to varying levels of dilution of a saline solution. PBS-(*n* = 5/time point) or mPEG (2 mM, 20 kDa; *n* = 5/time point)-treated RBCs were suspended in NaCl solution at increasing concentrations and incubated for 60 min at 37 °C. Erythrocyte hemolysis was initiated at about 0.55% NaCl for the control and mPEG-treated RBCS. The 50% erythrocyte hemolysis inflection points for the control and mPEG-treated RBCs were about 0.50% and 0.50%, respectively. The 100% erythrocyte hemolysis inflection points for the control and mPEG-treated RBCs were <0.35%. The control and mPEG-treated RBCs revealed no significant difference in their osmotic fragility curves for erythrocytes as a function of NaCl concentration (two-way ANOVA; group F_(1,208)_ = 0.194, *p* = 0.66; interaction F_(12,208)_ = 0.927, *p* = 0.521; Figure 4B).

### 3.4. Effect of PEGylation of Antigen A on Opsonized Erythrocyte Phagocytosis

To investigate the impact of mPEG treatment on the Fc receptor (FcR)-mediated phagocytosis of erythrocytes, we conducted a monocyte monolayer assay. Feline A-type red blood cells (RBCs) were coated with 20 kDa mPEG at a concentration of 2 mM, and after 1 h of incubation with feline anti-A alloantibodies, they were placed onto a monocyte layer and incubated for 2 h. The results revealed that mPEG-treated RBCs exhibited a significant inhibition of phagocytosis by monocytes compared to the control group. (* *p* < 0.001; Figure 5A,B).

## 4. Discussion

The present study showed, using gross agglutination and the microcolumn gel card method, that coating feline A-type RBCs with mPEG of 20 kDa and 2 mM completely blocked hemagglutination to feline anti-A alloantibodies over 8 h. Morphological examination revealed no difference in RBC size and shape between intact and mPEG-treated RBCs. An immunofluorescence study also confirmed that coating RBCs with mPEG inhibited the binding of feline anti-A alloantibodies. mPEG treatment on RBCs did not cause spontaneous hemolysis or osmotic fragility compared to control RBCs. In a monocyte monolayer assay, mPEG treatment significantly reduced feline anti-A antibody-mediated phagocystosis of RBCs. Taken together, these results demonstrate the feasibility of generating universal erythrocytes suitable for transfusion to B-type cats by manipulating feline A-type RBC with activated mPEG.

There have been studies to eliminate the antigenicity of incompatible bloods or xeno-RBCs. Scott et al. found that mPEG-treated human RBCs lose ABO blood group reactivity and decrease anti-blood group antibody binding [11]. Polymer-based coatings that are biocompatible with donor RBCs have been proposed to create stealth RBCs. Universal RBCs are constructed by applying polymers such as mPEG [17], polyoxazolines [18], and hyperbranched polyglycerols [19] to membrane proteins on the surface of allogeneic donor RBCs. It is known that the chemically activated polymers are covalently cross-linked to the exposed lysine residues of donor blood group antigens leading to the masking of antigenic recognition sites [10]. Many studies have focused on mPEG as the polymer of choice due to its superior ability to accomplish the safety and masking of allogeneic RBC [10,20,21]. mPEG, a compound of repeated HO-(CH_2_CH_2_O), is highly safe and has been approved by U.S FDA for oral, intravenous, subcutaneous, and intramuscular administration [22]. In the present study, while the 2, 5, and 10 kDa polymer sizes revealed positive agglutination, the 20 kDa polymer was highly effective in reducing agglutination scores, antibody binding, and antibody-mediated phagocytosis. Additionally, the immuneprotective concentrations of mPEG (20 kDa) did not alter the RBCs’ morphology nor normal physiochemical properties, as shown in osmotic fragility and spontaneous hemolysis tests. Consistent with our findings, previous studies showed that coating human Rh D+ RBCs with mPEG polymers of 2, 5 and 10 kDa is ineffective in preventing immune recognition and erythrophagocytosis, and that long-chain polymers of 20 and 30 kDa reduced phagocytosis in a monocyte monolayer assay [23]. Furthermore, mPEG treatment does not impair morphology, membrane topography, viability, or dynamics, as evidenced by normal oxygen uptake, delivery, and ion transport [17]. High concentration of PEG may affect RBC deformability [24]. However, a previous study also confirmed that RBC deformability is not altered by the mPEG coating at dosages of 0.6 mM–2.4 mM [17], which effectively blocked antigenic recognition and immunogenicity in the present study. It has been reported that the antiphagocytic effects of the 20 kDa polymer can be attributed to the loss of electrophoretic mobility of the modified RBC, thereby leading to impairment of the multifocal cell-to-cell interactions necessary for phagocytosis [25]. The present study found that coating of feline A-type RBC with long-chain mPEG (20–40 kDa) inhibited the binding of feline anti-A alloantibodies to A-type RBCs, as assessed by an immunofluorescence assay, and blocked hemagglutination to feline anti-A alloantibodies in a dose-dependent manner. This suggests that mPEG is covalently grafted to the protein of feline A-antigenic recognition sites and inhibits the antigenicity of feline A-type RBCs. On the other hand, while previous studies have shown the successful fabrication of human RBCs with mPEG of 20–30 kDa [12], the present study revealed that feline RBCs could be fabricated successfully using mPEG with molecular weights of 10–40 kDa. This suggests the fabrication conditions of mPEG for RBCs may be different among species. This is the first study in veterinary medicine showing successful fabrication of the RBC surface for universal bloods in companion animals.

The monocyte monolayer assay (MMA) is an in vitro assay used to examine the interaction of monocytes, RBCs and antibodies, and originally developed for predicting blood transfusion reactions in patients [26]. As the phagocytosis levels of RBCs in the MMA are closely correlated with the clinical outcomes of hemolysis, the MMA has been shown to have an excellent capacity for predicting which donor RBCs will survive normally in patients. This assay has been used as the sole method to select compatible blood units for transfusion in the difficult patients with auto- and allo-antibodies [27]. In the present study, uncoated RBCs (positive control) showed increased ‘rosette’ formation, in which RBCs bind to the periphery of the monocytes, which was significantly lowered at phagocytic index values of <5% by 20 kDa mPEG treatment at a concentration of 2 mM. Our findings, which showed that mPEG treatment significantly reduced feline anti-A antibody-mediated phagocytosis of RBCs, support the potential use of mPEG-treated A-type cat RBCs to attenuate acute transfusion reactions in type-B cats with anti-A alloantibodies, in situations wherein B-type cat RBCs are not available. While our study revealed promising in vitro data, in vivo animal studies demonstrating the survival rates of mPEG-treated cat RBCs were not performed due to the non-availability of B-type cats in a laboratory setting in Korea, which is a major limitation of our study.

In the present study, hemolysis was observed in mPEG of 30 and 40 kDa. To our knowledge, there is no report regarding the hemolytic activity of SVA-mPEG with large molecules. Although its mechanism is not clear, cytotoxicity (including cellular vacuolation) for PEG has been reported in laboratory animals. In particular, PEG-related cellular vacuolation has been observed more frequently with PEG of molecular weights > 30 kDa. We assume that the RBC hemolysis observed in mPEG of 30 and 40 kDa in the present study is associated with the cytotoxic effects of mPEG.

In conclusion, mPEG treatment of feline type-A RBCs preferentially inhibits hemagglutination to alloantibodies and anti-A antibody-mediated phagocytosis without changing the physiochemical properties of modified RBCs. It supports the potential use of mPEG-treated A-type cat RBCs to reduce the risk of acute transfusion reactions in type-B cats, in situations wherein B-type RBCs are not available.

## Figures and Tables

**Figure 1 jfb-14-00476-f001:**
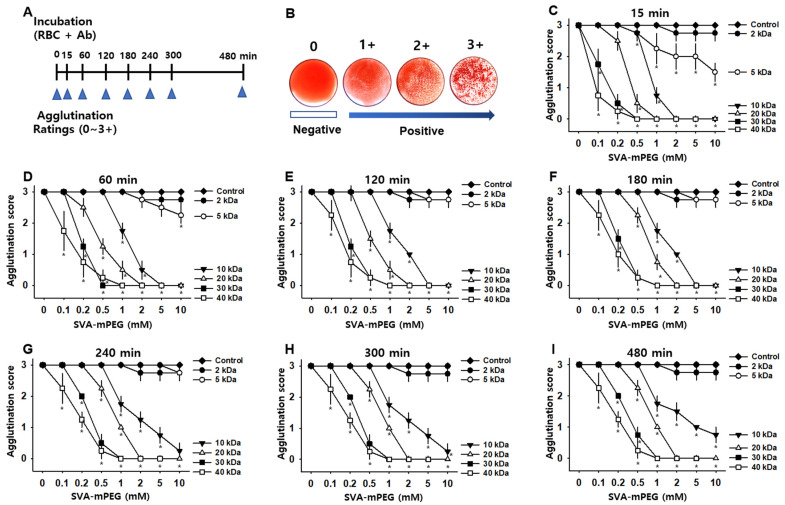
Effect of mPEG on agglutination of feline A-type RBCs. (**A**). Experimental schedule. Control and mPEG-treated RBCs were incubated with feline anti-A alloantibodies for 15, 60, 120, 180, 240, 300, or 480 min and then rated for agglutination. (**B**). Agglutination grading score: Agglutination intensity was graded on a scale from 0 to 3+: 0 (absence of agglutination), 1+ (mild agglutination), 2+ (moderate agglutination), and 3+ (strong agglutination). (**C**–**I**). Effect of mPEG on agglutination intensity. mPEGs of various molecular weights (2, 5, 10, 20, 30 and 40 kDa) and concentrations (0.1, 0.2, 0.5, 1, 2, 5 or 10 mM) were used at different time points (*n* = 4 samples/time point). * *p* < 0.05 vs. Control.

**Figure 2 jfb-14-00476-f002:**
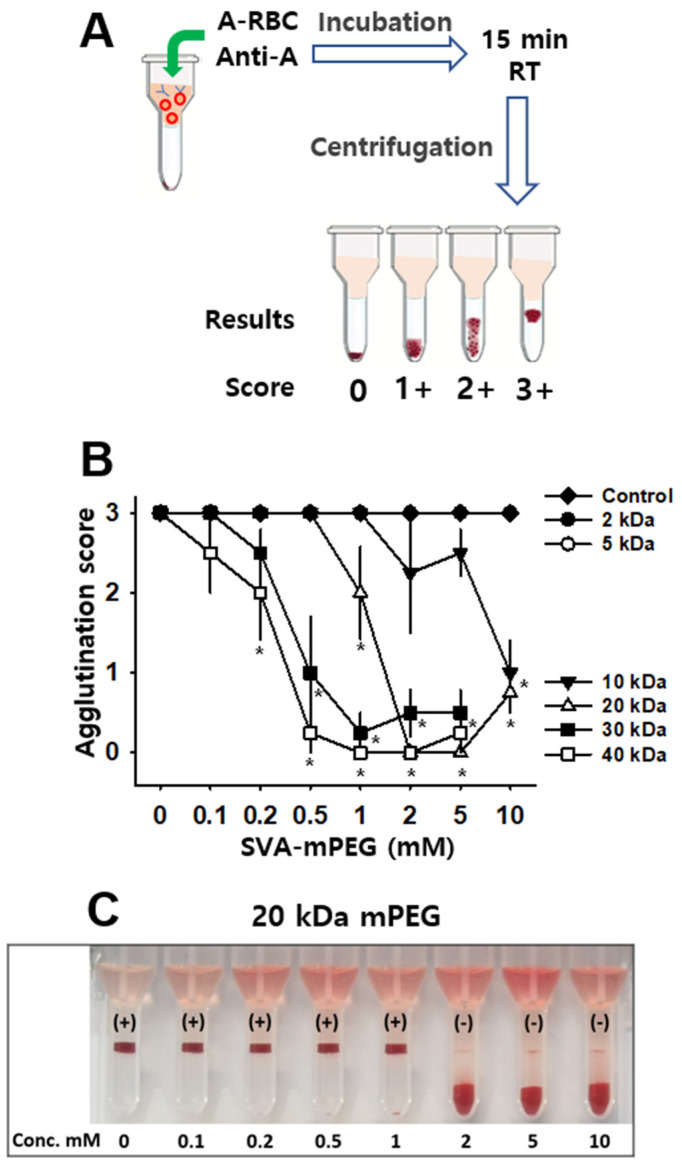
Effect of mPEG on microcolumn-based agglutination. (**A**). Schematics for microcolumn gel card method and agglutination grading scale for RBC retention in the gel: 0 (absence of agglutination or (-)), 1+ (mild agglutination), 2+ (moderate agglutination), and 3+ (strong agglutination). (**B**,**C**). Effect of mPEG on microcolumn-based agglutination. The mPEGs of different molecular weights (2, 5, 10, 20, 30 and 40 kDa) and concentrations (0.1, 0.2, 0.5, 1, 2, 5 or 10 mM) were used. *n* = 4 samples/point. *****
*p* < 0.05 vs. Control.

**Figure 3 jfb-14-00476-f003:**
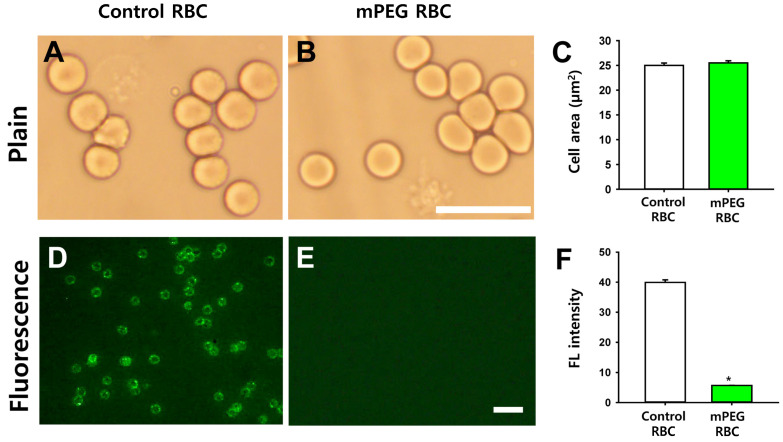
Morphology and antibody binding of mPEG-coated RBCs. (**A**,**B**). Representative bright field images of control (**A**) and mPEG-treated (**B**) feline A-type RBCs. Scale bar = 10 μm. (**C**) Mean cell size of control (*n* = 100) and mPEG-treated (*n* = 100) RBCs. *p* = 0.217. (**D**,**E**) Antibody binding inhibition via mPEG treatment of feline A-type RBCs. Fluorescence microscopy images of feline A-type RBCs coated with mPEG (**D**) or control RBCs (**E**). The cells were incubated with primary feline anti-A alloantibodies and secondary goat anti-feline IgG-FITC antibodies. Scale bar = 10 μm. (**F**) Fluorescence intensity of control (*n* = 54) and mPEG-treated (*n* = 54) RBCs. * *p* < 0.001.

**Figure 4 jfb-14-00476-f004:**
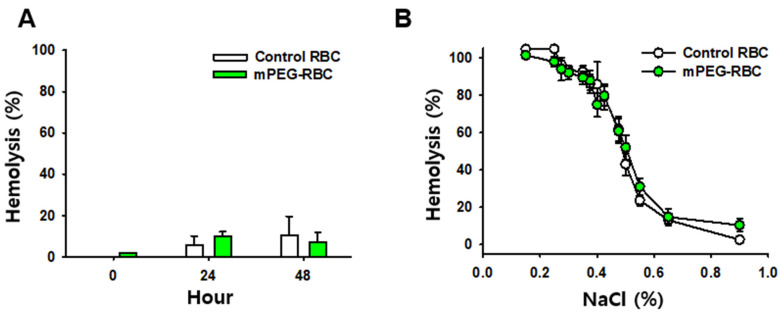
Physiochemical properties of mPEG-treated RBCs. (**A**). Assay of spontaneous hemolysis of control (*n* = 3/group) and mPEG-treated (*n* = 3/group) RBCs incubated for 0, 24 or 48 h. (**B**). Osmotic fragility of control (intact) or mPEG-treated (*n* = 9 samples/point) RBCs in different concentrations of NaCl solution.

**Figure 5 jfb-14-00476-f005:**
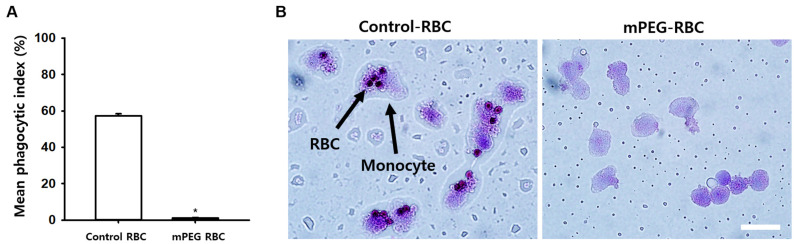
Effect of mPEG treatment on opsonized erythrocyte phagocytosis. (**A**). Monocyte monolayer assay showing the inhibition of phagocytosis by monocytes in mPEG-treated RBCs (*n* = 1470) compared to the control group (*n* = 325). * *p* < 0.001. (**B**). Representative phagocytic images of the control and mPEG-treated RBCs. Scale bar = 10 μm.

## Data Availability

The data presented in this study are available from the corresponding author upon reasonable request.

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
