# Peer review of "Production of Feline Universal Erythrocytes with Methoxy Polyethylene Glycol"

_jfb, 2023, doi:10.3390/jfb14090476_

Round 1
Reviewer 1 Report
The fabrication of universal erythrocytes is still a challenge which is not solved. The investagations are similar done by Scott in the 90s with human red blood cells. What are the mean differences in compatison to previous studies? Did you modify PEG? At least for humans it is known that more than 70 % of the population in the industrial countries have antibodies against PEG. It would be interesting to know if cats shows the same.
The proof of principle is missing. As long as it cannot be shown in animal studies that the PEG-RBCs do not lead to an immune response and are removed from the circulation, all in vitro results are nice but not convincing.
One important point has not been considered. The deformability of PEG-RBC will be different compared to normal RBC. This can easily determined by measurements of the whole blood apparent viscosity at high shear rates.
Some questions:
Composition of CPDA-1? Storage temperature? Is the gel card test validated for feline blood? Did you vary the ionic strength for the agglutination test?
2.3. what means normal RBC?
2.8. Positive control to test the function of monocytes?
3.2. How many cells were analysed? Maybe flowcytometry would be a better choice.
Author Response
Reviewer 1.
Q1) “The fabrication of universal erythrocytes is still a challenge which is not solved. The investagations are similar done by Scott in the 90s with human red blood cells. What are the mean differences in compatison to previous studies? Did you modify PEG? At least for humans it is known that more than 70 % of the population in the industrial countries have antibodies against PEG. It would be interesting to know if cats shows the same.”
Response: While the previous study done by Scott (Scott et al. 2017) showed successful fabrication of human RBCs with mPEG of 20-30 kDa, the present study revealed that feline universal RBCs could be produced successfully by using mPEG of 10-40 kDa and different dosages from human RBCs. It suggests the fabrication conditions of mPEG for RBCs may be different among species. It is also the first study in veterinary medicine showing successful fabrication of RBC surface for universal bloods in companion animals. It is described in Page 9, lines 324-330.
It has not been reported that PEG is immunogenic in cats. However, it is well-known that repeated injection of PEG is accepted for feline and has been widely used as a feline medication (Pedersen et al., J Feline Medicine and Surgery, 2019; Coleman et al, Vet J, 2014; Fiona et al., J Feline Med Surg. 2011; Friman et al., J Hepatol. 1990). As the side effects by systemic injection of PEG in feline have not been reported, it is not likely that PEG is highly immunogenic in cats.
Q2) “The proof of principle is missing. As long as it cannot be shown in animal studies that the PEG-RBCs do not lead to an immune response and are removed from the circulation, all in vitro results are nice but not convincing.”
Response: We absolutely agree with reviewer. As described in introduction, while A-typed cats are most common, B-typed cats are quite rare. B-typed cats are 1 % of feline population in Korea. In particular, Korean short-hair cats who approved by IACUC as experimental animals are almost all A-type. Because it is not possible to get B-typed cats in a laboratory setting, but not clinical cases, in Korea, we did not perform the in vivo experiment. It is a limitation of our study. Thus, we described a limitation inherent in our study in Page 10, lines 345-347.
Q3) “One important point has not been considered. The deformability of PEG-RBC will be different compared to normal RBC. This can easily determined by measurements of the whole blood apparent viscosity at high shear rates. “
Response: As reviewer comments, high concentration of PEG may affect RBC deformability (Mosbah et al,. Transplant Proc. 2006). However, a previous study confirmed that RBC deformability is not altered by the mPEG coating at dosages of 0.6 mM-2.4 mM, which effectively block antigenic recognition and immunogenicity in the present study. It is described in Page 9, lines 313-316.
Murad KL, Mahany KL, Brugnara C, Kuypers FA, Eaton JW, Scott MD. Structural and functional consequences of antigenic modulation of red blood cells with methoxypoly(ethylene glycol). Blood. 1999 Mar 15;93(6):2121-7.
Mosbah IB, Franco-Gou R, Abdennebi HB, Hernandez R, Escolar G, Saidane D, Rosello-Catafau J, Peralta C. Effects of polyethylene glycol and hydroxyethyl starch in University of Wisconsin preservation solution on human red blood cell aggregation and viscosity. Transplant Proc. 2006 Jun;38(5):1229-35.
Q4) Some questions:
Composition of CPDA-1? Storage temperature? Is the gel card test validated for feline blood? Did you vary the ionic strength for the agglutination test?
Response:
- CPDA-1 consists of dextrose, sodium citrate, citric acid, monobasic sodium phosphate and adenine and the bloods in CPDA-1 were stored at 1~6 °C until used. It is described in Page 2, lines 75-77.
- Gel card test has been utilized for feline blood (Spada et al., Front. Vet. Sci, 2020). We added the reference.
- As described in Page 2, lines 93-94, we used saline and did not vary the ionic strength for the agglutination test.
2.3. what means normal RBC?
Response: Normal RBC means untreated feline RBC. The term is revised as ‘feline RBC’ in Page 2, line 82.
2.8. Positive control to test the function of monocytes?
Response: Positive control shows that Fc receptor (FcR)-mediated phagocytosis of erythrocytes by monocytes normally functions in our experimental setting. It is explained in Page 10, lines 338-339.
3.2. How many cells were analysed? Maybe flowcytometry would be a better choice.
Response: In Fig. 3, we analyzed fluorescence intensity of control (n=54 cells) and mPEG-treated (n=54 cells) RBCs. As reviewer comments, we tried flowcytometry but failed to get meaningful data. The reason is that the concentration of feline alloantibody used (1:8 titre) agglutinated blood cells moderately and it hindered the analysis of flowcytometry.

Reviewer 2 Report
This paper deals with the effect of pegylation of feline red blood cells with PEGs of different molecular weight and concentration, and its effect on the properties of such conjugates. The paper is recommended to be accepted for publication after some revision on the basis of comments below.
COMMENTS
1.
The following publications are suggested to be studied by the authors, mentioned in the text of this paper and cited in the References:
Gavazza, A.; Rossi, G.; Antognoni, M. T.; Cerquetella, M.; Miglio, A.; Mangiaterra, S. Feline blood groups: A systematic review of phylogenetic and geographical origin. Animals 2021, 11, 3339.
Spada, E.; Carrera Nulla, A.; Perego, R.; Baggiani, L.; Proverbio, D. Evaluation of Association between Blood Phenotypes A, B and AB and Feline Coronavirus Infection in Cats. Pathogens 2022, 11, 917.
Koenig, A.; Maglaras, C. H.; Giger, U. Acute hemolytic reaction due to A‐B mismatched transfusion in a cat with transient AB blood type. J. Vet. Emerg. Crit. Care 2020, 30, 325-330.
2.
The authors should describe at least briefly which kind of chemical bonding occurs between the applied methoxy-PEG succinimidyl valerate (mPEG-SVA) and the red blood cells (RBCs).
3.
The efficiency of the treatment of the RBCs with the mPEG-SVAs should be presented.
4.
A summarizing table for the results of the effect of mPEG-SVA treatment on the antigenic recognition is suggested in order to present much better way of the obtained results.
5.
The authors should provide detailed explanation why the agglutination increases with increasing mPEG-SVAs in higher concentrations (see Figure 2B).
6.
The format of the list of references does not follow the requirement of the journal. This should be corrected by the authors.
7.
Reference 20 in the References section is incomplete (year, volume, page numbers are missing).
Author Response
Reviewer 3.
The paper is recommended to be accepted for publication after some revision on the basis of comments below.
- The following publications are suggested to be studied by the authors, mentioned in the text of this paper and cited in the References:
Gavazza, A.; Rossi, G.; Antognoni, M. T.; Cerquetella, M.; Miglio, A.; Mangiaterra, S. Feline blood groups: A systematic review of phylogenetic and geographical origin. Animals 2021, 11, 3339.
Spada, E.; Carrera Nulla, A.; Perego, R.; Baggiani, L.; Proverbio, D. Evaluation of Association between Blood Phenotypes A, B and AB and Feline Coronavirus Infection in Cats. Pathogens 2022, 11, 917.
Koenig, A.; Maglaras, C. H.; Giger, U. Acute hemolytic reaction due to A‐B mismatched transfusion in a cat with transient AB blood type. J. Vet. Emerg. Crit. Care 2020, 30, 325-330.
Response: The above papers are mentioned and cited in Page 1, lines 39 and 45.
- The authors should describe at least briefly which kind of chemical bonding occurs between the applied methoxy-PEG succinimidyl valerate (mPEG-SVA) and the red blood cells (RBCs).
Response: It is known that the chemically activated polymers are covalently cross-linked to the ex-posed lysine residues of donor blood group antigens leading to masking antigenic recog-nition sites (Garratty 2004). It is described in Page 9, lines 295-298.
- The efficiency of the treatment of the RBCs with the mPEG-SVAs should be presented.
Response: While our study revealed promising in vitro data, in vivo animal studies demonstrating survival rates of mPEG-treated cat RBCs were not performed. As described in introduction, while A-typed cats are most common, B-typed cats are quite rare. B-typed cats are 1 % of feline population in Korea. In particular, Korean short-hair cats who approved by IACUC as experimental animals are almost all A-type. Because it is not possible to get B-typed cats in a laboratory setting, but not clinical cases, in Korea, we did not perform the in vivo experiment. It is a limitation of our study. Thus, we described a limitation inherent in our study in Page 10, lines 345-347.
- A summarizing table for the results of the effect of mPEG-SVA treatment on the antigenic recognition is suggested in order to present much better way of the obtained results.
Response: We agree with reviewer’s comment that figure 1 is complicated to understand. As Figure 1 is most important through this study, we believe that raw data of figure 1 itself would provide an important information for investigators. Instead of the table, we summarized figure 1 in Page 5, lines 199-201.
- The authors should provide detailed explanation why the agglutination increases with increasing mPEG-SVAs in higher concentrations (see Figure 2B).
Response: In Fig. 2B, one of 4 samples at 10 mM displayed 2+ agglutination, maybe due to a technical error and it caused increased agglutination scores than that of 5 mM. It is described in Page 5, lines 198-199.
- The format of the list of references does not follow the requirement of the journal. This should be corrected by the authors.
Response: Sorry for this issue. I am on attending IBRO meeting in Spain and cannot access Endnote program. Because of short revision due date, the issue would be corrected simply on the stage of publications by using Endnote program. Please understand this situation.
- Reference 20 in the References section is incomplete (year, volume, page numbers are missing).
Response: It is corrected.

Reviewer 3 Report
In this paper, the authors described an effective approach that employs polyethylene glycol-modified feline red blood cells as universal erythrocytes for transfusion to B-typed cats. The effect on the molecular weights of PEG was explored and it was found that the 20 kDa PEG could block hemagglutination to feline anti-A alloantibodies for an extended period of time. The overall research is very interesting and the method proposed could be important in veterinary clinical practice. I would recommend the publication of this paper if the authors could address the following comments:
1. Why does the 30 and 40 kDa mPEG lead to slight hemolysis? What is the rationale for choosing the molecular weights presented in the method?
2. The arrangement of Figure 1 can be improved. I suggest the authors summarize Figure 1 C-I into a table and put these into supplementary figures.
3. Does PEGylation of the cell surface affect the cells’ viability?
4. The current reference is too thin. The authors can do a better job.
Author Response
Reviewer
I would recommend the publication of this paper if the authors could address the following comments:
- Why does the 30 and 40 kDa mPEG lead to slight hemolysis? What is the rationale for choosing the molecular weights presented in the method?
Response: To our knowledge, there is no report regarding hemolytic activity by SVA-mPEG of large molecules. Although the mechanism is not clear, cytotoxicity including cellular vacuolation for PEG has been reported in laboratory animals. In particular, PEG-related cellular vacuolation has been observed more frequently at PEG of molecular weights >30 kDa. We assume that RBC hemolysis observed in mPEG of 30 and 40 kDa in the present study is associated with cytotoxic effects of PEG. It is incorporated into Page 10, lines 348-354.
The molecular weights of mPEG were chosen based on a previous human RBC study. Scott et al tested 2-30 kDa mPEG for human RBC and reported successful production of human stealth RBCs with mPEG of 20 and 30 kDa. It is described in Page 2, lines 83-84.
- The arrangement of Figure 1 can be improved. I suggest the authors summarize Figure 1 C-I into a table and put these into supplementary figures.
Response: We agree with reviewer’s comment that figure 1 is complicated to understand. As Figure 1 is most important through this study, we believe that raw data of figure 1 itself would provide an important information for investigators. Instead of the table, we summarized figure 1 in Page 5, lines 199-201.
- Does PEGylation of the cell surface affect the cells’ viability?
Response: It is reported that the mPEG treatment does not impair viability, morphology, membrane topography or dynamics as evidenced by normal oxygen uptake and delivery, and ion transport (Murad et al. 1999). High concentration of PEG may affect RBC deformability (Mosbah et al. 2006). However, a previous study also confirmed that RBC deformability is not altered by the mPEG coating at dosages of 0.6 mM-2.4 mM, which effectively block antigenic recognition and immunogenicity (Murad et al. 1999). It is described in Page 9, lines 311-316.
- The current reference is too thin. The authors can do a better job.
Response: In responses to both reviewers’ comments, we added more references.

Round 2
Reviewer 1 Report
My comments have been considered.